# Prognostic Impact of Pretreatment 2-[^18^F]-FDG PET/CT Parameters in Primary Gastric DLBCL

**DOI:** 10.3390/medicina57050498

**Published:** 2021-05-14

**Authors:** Domenico Albano, Francesco Dondi, Angelica Mazzoletti, Pietro Bellini, Raffaele Giubbini, Francesco Bertagna

**Affiliations:** Nuclear Medicine Department, University of Brescia and ASST Spedali Civili of Brescia, 25123 Brescia, Italy; f.dondi@outlook.it (F.D.); mazzolettiangelica@gmail.com (A.M.); bellini.pietro@outlook.it (P.B.); raffaele.giubbini@unibs.it (R.G.); francesco.bertagna@unibs.it (F.B.)

**Keywords:** primary gastric DLBCL, 2-[^18^F]-FDG PET/CT, MTV, TLG, diffuse large B cell lymphoma, prognosis

## Abstract

*Background and Objectives*: Primary gastric diffuse large-B cell lymphoma (DLBCL) is an aggressive lymphoma subtype with high ^18^F-FDG avidity but unclear criteria for 2-[^18^F]-FDG PET/CT in the evaluation of treatment response and prognostication. Our aim was to investigate whether the pretreatment 2-[^18^F]-FDG PET/CT variables may predict treatment response (at end of first-line therapy) and prognosis in primary gastric DLBCL. *Materials and Methods*: we included 57 patients with a diagnosis of primary gastric DLBCL and a baseline 2-[^18^F]-FDG PET/CT and an end of treatment PET/CT after 6 cycles of R-CHOP chemotherapy. We analyzed PET images qualitatively and semi-quantitatively by deriving the maximum standardized uptake value body weight (SUVbw), the maximum standardized uptake value lean body mass (SUVlbm), the maximum standardized uptake value body surface area (SUVbsa), lesion to liver SUVmax ratio (L-L SUV R), lesion to blood-pool SUVmax ratio (L-BP SUV R), metabolic tumor volume and total lesion glycolysis of gastric lesion (gMTV and gTLG), and total MTV (tMTV) and TLG. Survival curves were plotted according to the Kaplan–Meier analysis. *Results:* at a median follow up of 80 months, the median PFS and OS were 69 and 80 months. Baseline gMTV, gTLG, tMTV, and TLG were significantly higher in patients with incomplete response (partial response and progression) compared to complete response group. tMTV and TLG were confirmed to be independent prognostic factors both for PFS (*p* = 0.023 and *p* = 0.038) and OS (*p* = 0.038 and *p* = 0.026); instead, the other metabolic parameters were not related to outcome survival. *Conclusions*: high tMTV and TLG were significantly correlated with shorter survival (PFS and OS) and may predict incomplete response after therapy.

## 1. Introduction

Primary gastric lymphoma (PGL) is a rare lymphoma, and among them, diffuse large B-cell lymphoma (DLBCL) variant accounts for about 59% of cases [1]. Nowadays, chemotherapy regimens, such as the combination of cyclophosphamide, doxorubicin, vincristine, and prednisone (CHOP) and rituximab (R-CHOP), have been considered as the front-line therapy for primary gastric DLBCL [2]. The treatment response and the outcome of patients affected by primary gastric DLBCL have been reported to be different and heterogeneous, according to histological and clinical features [1]. The potential usefulness of fluorine-18-fluorodeoxyglucose positron emission tomography/CT (2-[^18^F]-FDG-PET/CT) in the study of Hodgkin’s lymphoma (HL) and aggressive NHL has been widely validated both in staging, restaging, and treatment response fields [3,4]. However, the usefulness of 2-[^18^F]-FDG-PET/CT in primary gastric lymphoma is unclear and directly related to the histotype of lymphoma. For gastric mucosa associated lymphoid tissue (MALT) lymphoma, the FDG-avidity rate is widely variable and correlated with morphological and histological features [5,6]; while for gastric DLBCL the PET/CT detection rate is optimal [7,8,9]. Moreover, considering the metabolic behavior of DLBCL at 2-[^18^F]-FDG-PET/CT, this tool and their metabolic parameters are recommended for prognostic assessment and management in DLBCL [10,11]; however strong evidences about the prognostic role of semiquantitative 2-[^18^F]-FDG-PET/CT variables in primary gastric DLBCL are lacking, with only promising findings [12,13,14].

The aim of our study was to investigate the potential role of pretreatment PET/CT parameters in predicting treatment response (at end of treatment) and prognosis in patients affected by primary gastric DLBCL.

## 2. Materials and Methods

### 2.1. Patients Features

Between February 2009 and June 2020, 57 patients with a diagnosis of primary gastric DLBCL were retrospectively included. We revised the medical data and pathology reports of these patients: the main epidemiological (age at diagnosis, gender, Ann Arbor stage system), morphological (localization of primary disease, tumor size, extranodal or nodal localization, presence of gastritis, ulcer, Helicobacter pylori), and clinical features (presence of B symptoms, IPI score, LDH level, Ki-67 score) and PET/CT parameters, type of therapy, and follow-up data were collected. Lactate-dehydrogenase (LDH) level and International prognostic index (IPI) score were dichotomized using a cutoff value of 245 U/L and 2, respectively. According to Ann Arbor stage, we divided our population in early-stage (I and II) and advanced-stage (III and IV). All patients were treated with chemotherapy regimen using 6 cycles of R-CHOP.

### 2.2. 2-[^18^F]-FDG-PET/CT Imaging and Interpretation

Patients performed a pretreatment 2-[^18^F]-FDG-PET/CT scan before any therapy and a further 2-[^18^F]-FDG-PET/CT at the end of first-line treatment (after six cycles of chemotherapy). 2-[^18^F]-FDG-PET/CT scans were performed after at least 6 h of fasting and with glucose level lower than 150 mg/dl. About 60 min after an intravenously injection of the radiotracer (activity of 3.5–4.5 MBq/Kg), PET/CT scans were acquired from the skull basis to the mid-thigh on a Discovery ST PET/CT tomograph (General Electric Company—GE^®^—Milwaukee, WI, USA) with conventional parameters (CT: 80 mA, 120 Kv without contrast; 2.5–3.5 min per bed-PET-step of 15 cm); the matrix reconstruction was 256 × 256 and the field of view 60 cm. During the uptake time, the patients were invited to drink water; no contrast agents (oral or intravenous) were used; written consent was obtained before studies. Pre-treatment 2-[^18^F]-FDG-PET/CT scans were done within 7 days before starting chemotherapy, while the end of treatment 2-[^18^F]-FDG-PET/CT were acquired at least 21 days after the completion of chemotherapy.

The PET images were analyzed by two nuclear medicine physicians with experience in PET/CT and lymphoma (F.B, D.A) qualitatively and semi-quantitatively by measuring some absolute parameters: the maximum standardized uptake value body weight (SUVbw), maximum standardized uptake value lean body mass (SUVlbm), maximum standardized uptake value body surface area (SUVbsa); some ratios: lesion to liver SUVmax ratio (L-L SUV R) and lesion to blood-pool SUVmax ratio (L-BP SUV R); some tumor burden variables: gastric metabolic tumor volume (gMTV) and gastric total lesion glycolysis (gTLG) of the gastric lesion. Moreover, also total MTV (tMTV) as the sum of MTV of all hypermetabolic lesion and corresponding TLG were calculated.

For the visual analysis, the readers had knowledge of anamnesis, and described as suspected of disease every focal FDG uptake different from physiological distribution and background. For the measurement of the SUV of the lesion with higher uptake, they draw a region of interest (ROI) over the area of maximum uptake in the stomach and they calculated the SUVmax as the highest SUV of the pixels within the ROI. Using a round-shape ROI of 1 cm of diameter at the VIII hepatic segment of transaxial PET images, they calculated the hepatic SUVmax; with a similar ROI put at the aortic arch avoiding the vessel wall they calculated the blood-pool SUVmax. The FDG-positive gastric lesion was taken as reference lesion in each patient and the SUVbw, SUVlbm, SUVbsa, L-L SUV R, L-BP SUV R of that lesion were derived. With the help of a SUV-based automated contouring tool (Advantage Workstation 4.6, GE HealthCare), they calculated gMTV and tMTV; as recommended by European Association of Nuclear Medicine, the threshold method used was based on 41% of the SUVmax [15]. Subsequently, total MTV (tMTV) was obtained by the sum of all nodal and extranodal lesions. gTLG and TLG were calculated as the sum of the product of gMTV and its SUVmean and the product of MTV of each lesion and its SUVmean.

### 2.3. Statistical Analysis

For the statistical analysis we used the MedCalc Software version 18.1 (Ostend, Belgium). Categorical features were described as simple and relative frequencies; while the numeric features were represented as mean, standard deviation, median, and range.

The Kolmogorov–Smirnov test was performed to test the normality distribution and demonstrated that the normality distribution was not confirmed.

For the comparison of semiquantitative metabolic factors (SUVbw, SUVlbm, SUVbsa, L-L SUV R, L-BP SUV R, MTV, and TLG) between incomplete and complete metabolic response groups at the end of therapy we adopted the Mann–Whitney test.

For the whole population, the Youden index from the receiver operating characteristic (ROC) curve analysis was used to identify the best threshold point of semiquantitative variables in the light of which interpret the results of progression free survival (PFS). Treatment response was defined evaluating end of treatment 2-[^18^F]-FDG-PET/CT, as suggested by the Lugano classification [3,4]; Deauville score 1, 2, and 3 at the end of first-line treatment was judged negative scan. PFS was measured from the date of baseline 2-[^18^F]-FDG-PET/CT to the date of first recurrence, disease progression, death, or the date of the last follow-up. OS was measured from the date of baseline 2-[^18^F]-FDG-PET/CT to the date of death or to the date of last follow-up. After the binarization of continuous PET/CT variables, survival curves were also generated using a Kaplan–Meier (KM) approach and using a two-tailed log rank test was applied to investigate the differences between groups. The hazard ratio (HR) and its confidence interval (CI) were estimated by Cox regression. Martingale residuals and cox proportional hazard regression analysis were performed to test the assumption of proportionality of risks and the linearity. The data did not violate the proportional hazard assumption and the linearity was met. A *p* value of < 0.05 was considered as statistically significant.

## 3. Results

### 3.1. Patients’ Features

Among 57 patients with a histological diagnosis of primary gastric DLBCL, 32 (56%) were male and 25 (44%) female; median age was 66. According to the Ann Arbor system, 19 patients were at stage I, 12 at stage II, 8 at stage III, and 18 at stage IV.

Only in 16 cases B-symptoms were present; LDH level was higher than normal interval (245 U/L) in 23 patients and IPI score > 2 in 19 cases. Ki-67 score was high in most cases (Table 1). Pretreatment 2-[^18^F]-FDG-PET/CT scan was positive in all patients showing the presence of an increased FDG uptake corresponding to gastric lesion; antrum is the most frequent site of disease, followed by body. Median tumor size was 30 mm and concomitant gastritis, gastric ulcer, and H. Pylori infection were present in 71%, 72%, and 49%, respectively. Median SUVbw of the gastric lesion was 21.1; median SUVlbm was 15.4, median SUVbsa was 5.2, median L-L SUV R 8.3, median L-BP SUV R 11.4, median gMTV 24.1 cm^3^, gTLG 4435, tMTV 32.95 cm^3^, and median TLG was 578.5 (Figure 1).

### 3.2. Evaluation of Treatment Response

Considering the metabolic response categories, derived by Lugano classification [3,4], at the end of treatment 41 (72%) patients presented complete response while 14 (24.5%) patients presented partial metabolic response and 2 (3.5) progression of disease.

gMTV, gTLH, tMTV were significantly different between complete response and incomplete (partial and progression) metabolic response group (*p* = 0.019, *p* = 0.004, *p* = 0.013, and *p* = 0.004, respectively; Table 2). Instead, no significant differences were founded considering the remaining PET/CT variables.

### 3.3. Prognostic Role of 2-[^18^F]-FDG-PET/CT

At a median follow-up of 80 months, 26 patients had relapse or progression of disease with a mean time of 51 months (range: 1–152 months) from the pre-treatment PET/CT scan, while death happened in 21 patients with a mean time of 70 months (range 2–156). The median PFS was 69 months (range: 1–183 months) and the median OS was 80 months (range: 2–183 months). The estimated 3-year and 5-year PFS were 51% and 49%; the 3-year and 5-year OS were 60% and 59%.

In univariate analysis, the semiquantitative parameters (SUVbw, SUVlbm, SUVbsa, L-L SUV R, and L-BP SUV R) dichotomized applying ROC analyses (Table 3) were not associated with survival, both for PFS and OS. Instead, gMTV, gTLG, tMTV, and TLG were significantly correlated with prognosis (Figure 2 and Figure 3). The other clinical/pathological features (gender, age, tumor stage, B symptoms, IPI score, Ki-67 score, LDH level, and tumor size) did not correlate with outcome (Table 4). In multivariate analysis, tMTV and TLG were the only independent prognostic parameters for PFS (*p* = 0.023 and *p* = 0.038) and OS (*p* = 0.038 and *p* = 0.026) (Table 4).

In patients with a tMTV ≥119 cm^3^, survival rates were significantly worse in comparison with patients with lower tMTV; in fact, the 3-year PFS was 33% compared to 61% (*p* < 0.001) and the 3-year OS was 41% compared to 70% (*p* < 0.001). Besides, in patients with high TLG (≥852), survival rates were significantly shorter in comparison with those with low-TLG (a 3-year PFS of 40% and 3-year OS of 38% compared to 64% and 61%) (*p* < 0.001).

## 4. Discussion

Besides the standard visual analysis and qualitative criteria, such as Deauville criteria, there is increasing evidence of the usefulness of semiquantitative parameters from baseline 2-[^18^F]-FDG-PET/CT in patients with HL and aggressive NHL, especially in prognostic field [2,3]. The most famous and utilized semiquantitative PET/CT factor is SUV and it is generally accepted in the current published literature for assessing disease activity in lymphoma, due to the automatic and easy way to calculate. However, despite several positive results of SUV as prognostic factor in primary gastric DLBCL [16], SUV is for definition characterized by some factors that may affect their accuracy, like the uptake time between injection and imaging scan, the partial volume effect in case of small lesion, the risk of extravasation of administered radiotracer at the site of injection, the residual activity lost in the syringe, the decay of the injected activity and the technological features of the scanner, and acquisition and reconstruction parameters [17]. More recently, some variables indirectly expressing the volume-based metabolic disease, have emerged and seem to have a positive role in the evaluation of treatment response and outcome in different lymphoma [18,19,20,21,22,23,24]. These parameters are MTV and TLG; additionally, in primary gastric DLBCL, some findings are available [11,12,13,14], but shared consensus is lacking.

MTV and TLG are features that take into account both the size as well as the radiotracer uptake, reflecting a combination of morphological and functional aggressiveness.

In our study, we studied the potential prognostic role of several metabolic features (SUV-related variables and MTV and TLG) demonstrating that SUV value were not significantly correlated with prognosis (PFS, OS). Instead, total MTV and TLG showed to be independent prognostic factors both for PFS and OS. Patients with high tMTV and TLG showed a worse survival, both for PFS and OS, underlying that in cases of high volumes of disease the prognosis was reduced. MTV and TLG values of stomach lesions, showed to be significantly correlated with survival but only at univariate analysis.

This evidence confirmed previous papers that underlined the superiority of MTV and TLG in comparison with SUV values in prognostication field.

In this manuscript, we chose to investigate a high number of metabolic variables not previously investigated, such as SUV corrected for lean body mass, for body surface area, and some ratios (lesion to liver SUVmax ratio and lesion to blood pool SUVmax ratio) to understand if “not common” metabolic features may play a role or not.

Baseline total MTV and TLG were the most robust predictor of outcome in our analysis, even better than the main clinical, morphological, and histological features.

It seems interesting to note that metabolic volumetric parameters (MTV and TLG) have a prognostic value, independently from stage of disease. The thresholds proposed by our work were 119 cm^3^ for tMTV and 852 for TLG; however, it is premature to suggest these cut-off values as universal because are strictly dependent to patients features and center characteristics.

Our results are similar to Jiang et al. [14] and Zhao et al. [13] that demonstrated that MTV, TLG, and National Comprehensive Cancer Network International Prognostic Index (NCCN-IPI) significantly predicted the prognosis.

Of course, other papers based upon larger sample are shareable to support or deny our preliminary evidence.

2-[^18^F]-FDG-PET/CT radiomic features could play a role in predicting treatment response and prognosis in primary gastric DLBCL, as demonstrated by recent articles [25,26], but several methodological and technical doubts are present, such as the choice of features to include, their clinical meaning, the reproducibility of the results. However, also in this case, further studies with larger populations are needed to clarify the impact of texture analysis.

Another topic studied in this manuscript was the relationship between the baseline metabolic variables and treatment response. The detection of factors that may help to predict treatment response is crucial because it allows to personalize the management adjusting therapies according to class risk. In our study, we demonstrated a positive impact of gMTV, gTLG, tMTV, and TLG in predicting response after first line of therapy. In fact, these parameters were significantly higher in patients with incomplete response (including partial response and progression of disease according to Lugano stage) than complete response after the end of chemotherapy.

Also in this field, the metabolic burden disease expressed as MTV and TLG seems to have the more significant impact.

Nowadays, the application of metabolic tumor volumes (MTV and TLG) in the clinical routine seems to be premature, due to the lack of a standardized and shared methodology for their measurements and the heterogeneity of results between different centers. Different methods for the measurements of MTV and TLG are describe in literature and a wide interval of thresholds have been proposed to calculate these parameters. In this study, we used a method suggested by EANM [15] that chose an isocontour cutoff tool based on 41% of the SUVmax.

The effective role of MTV and TLG in risk stratification and the possibility to combine these variables with other clinical or imaging features should be evaluated in further studies and may increase the accuracy of this method.

Some limitations of this manuscript study are the retrospective nature and the relatively low number of patients investigated, due to the rarity of this disease.

## 5. Conclusions

In conclusion, with this study we demonstrated that MTV and TLG were the only two parameters significantly correlated with survival (PFS and OS) and with treatment response.

## Figures and Tables

**Figure 1 medicina-57-00498-f001:**
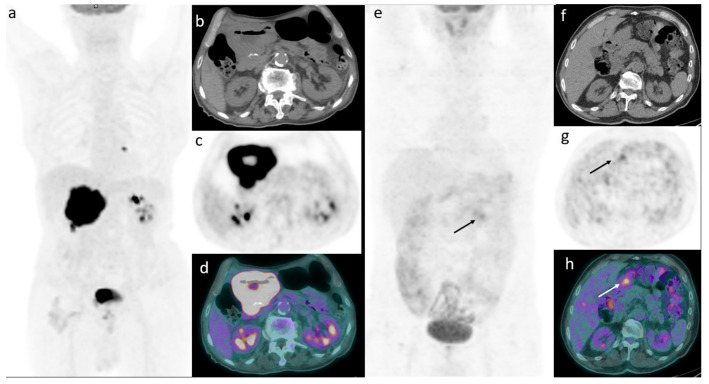
An example of a 76-year-old man affected by primary gastric DLBCL and a mediastinal lymph node. Baseline maximum intensity projection (MIP, **a**) revealing diffuse hypermetabolic stomach uptake involving whole gastric wall with a SUVbw of 30, SUVlbm 24.5, SUVbsa 8.1, L-L SUV R 9.5, L-BP SUV R 13.4, gMTV 196, and gTLG 4317. Transaxial CT (**b**), transaxial PET (**c**) and transaxial PET/CT (**d**) showing the increased diffuse gastric uptake. Another example in a 46-year-old male with a diagnosis of primary gastric DLBCL and no other localizations of disease. MIP (**e**) showing faint-moderate uptake in the abdomen corresponding to a focal increased uptake in the antrum with a SUVbw of 5, SUVlbm 4.2, SUVbsa 1.4, L-L SUV R 2.4, L-BP SUV R 2.1, gMTV 19.9, and gTLG 165. Transaxial CT (**f**), transaxial PET (**g**), and transaxial PET/CT (**h**) displaying the increased FDG uptake.

**Figure 2 medicina-57-00498-f002:**
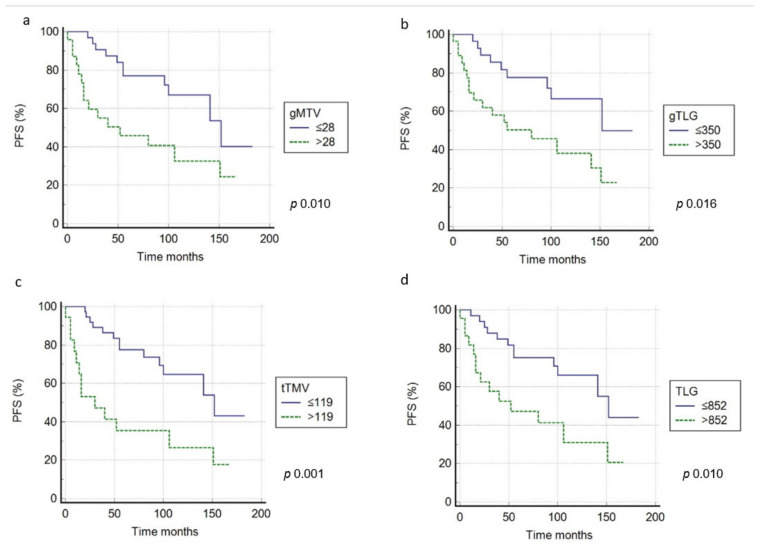
PFS curves according to pretreatment gMTV (**a**), gTLG (**b**), tMTV (**c**), and TLG (**d**) threshold.

**Figure 3 medicina-57-00498-f003:**
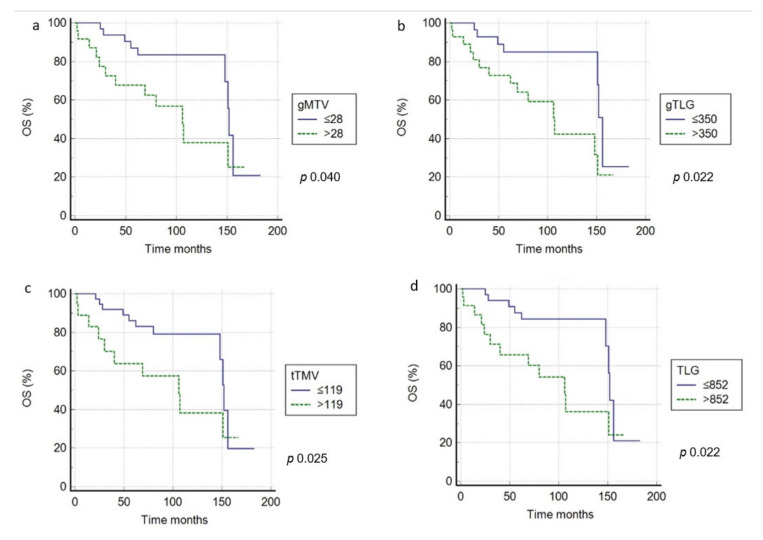
OS curves according to pretreatment gMTV (**a**), gTLG (**b**), tMTV (**c**), and TLG (**d**) threshold.

**Table 1 medicina-57-00498-t001:** The main features of our population.

	Patients *n* (%)
Age median (range)	66 (17–84)
Sex	
male	32 (56%)
female	25 (44%)
Ann Arbor stage at diagnosis	
I	19 (33%)
II	12 (21%)
III	8 (14%)
IV	18 (32%)
Presence of B symptoms	16 (28%)
Site of disease	
Fundus	3 (5%)
Body	10 (18%)
Antrus	28 (49%)
Cardias	3 (5%)
Diffuse	13 (23%)
Tumor size, median (range) mm	30 (9–153)
Gastritis	40 (71%)
Ulcer gastric	41 (72%)
Presence of Helicobacter pylori	28 (49%)
Ki-67 score ≤ 15%	10 (18%)
Ki-67 score > 15%	47 (82%)
LDH ≤ 245 U/L	34 (60%)
>245 U/L	23 (40%)
IPI score ≤ 2	38 (67%)
>2	19 (33%)
SUVbw median (range) g/mL	21.1 (3.6–62)
SUVlbm median (range) g/mL	15.4 (2.9–36)
SUVbsa median (range) cm^2^/mL	5.2 (1–11.3)
L-L SUV ratio median (range)	8.3 (1.3–21)
L-BP SUV ratio median (range)	11.4 (1.9–28.7)
gMTV median (range) cm^3^	24.1 (3.8–1100)
gTLG median (range)	4435 (22–18,313)
tMTV median (range) cm^3^	32.95 (4–12,500)
TLG median (range)	578.5 (22–29,670)

IPI: international prognostic score; LDH: lactate dehydrogenase.

**Table 2 medicina-57-00498-t002:** Comparison between different metabolic response groups at end-of-treatment considering the main PET/CT baseline features.

Variable (Median)	Metabolic Response at the End of Treatment		*p* Value
	Incomplete Response (*n* = 16)	Complete Response (*n* = 41)	
SUVbw g/mL	22	18.8	0.063
SUVlbm g/mL	16.1	14.5	0.512
SUVbsa cm^2^/mL	5.7	5.1	0.271
L-L SUV R	11.4	8.2	0.082
L-BP SUV R	12.2	10.1	0.126
gMTV cm^3^	225	72.9	0.019
gTLG	4290	1300	0.004
tMTVcm^3^	245	99	0.013
TLG	5701	1565	0.004

**Table 3 medicina-57-00498-t003:** Semiquantitative PET/CT variables thresholds derived from ROC analysis.

	ROC Analysis				
Variables	Threshold	AUC (95% CI)	*p* Value	Sensitivity (95% CI)	Specificity (95% CI)
SUVbw g/mL	17.1	0.565 (0.420–0.700)	0.410	71% (49–87)	50% (31–69)
SUVlbm g/mL	19.6	0.511 (0.371–0.650)	0.890	87.5% (68–97)	33% (17–53)
SUVbsa cm^2^/mL	3.6	0.553 (0.411–0.688)	0.508	87.5% (68–97)	33% (17–53)
L-L SUV R	10.8	0.578 (0.436–0.712)	0.322	46% (26–67)	73% (54–88)
L-BP SUV R	2.8	0.564 (0.422–0.698)	0.422	96% (79–100)	20% (8–39)
gMTV cm^3^	28	0.647 (0.505–0.772)	0.059	62.5% (41–82)	70% (51–85)
gTLG	350	0.656 (0.515–0.780)	0.039	71% (49–87)	63% (44–80)
tMTV cm^3^	119	0.694 (0.553–0.812)	0.008	54% (33–74)	83% (65–94)
TLG	852	0.680 (0.539–0.800)	0.014	58% (37–78)	73% (54–88)

CI: confidence interval; AUC: area under curve; SUV: standardized uptake value; bw: body weight; lbm: lean body mass; bsa: body surface area; L-L R: lesion to liver ratio; L-BP R: lesion to blood pool ratio; gMTV: gastric metabolic tumor volume; TLG: total lesion glycolysis.

**Table 4 medicina-57-00498-t004:** Univariate and multivariate analyses for PFS and OS.

	Univariate Analysis		Multivariate Analysis	
	*p* Value	HR (95% CI)	*p* Value	HR (95% CI)
PFS				
Gender	0.651	1.197 (0.547–2.620)		
Age years	0.688	0.819 (0.313–2.166)		
Tumor Stage	0.229	1.614 (0.739–3.522)		
B symptoms	0.673	1.218 (0.479–3.121)		
IPI score	0.882	0.939 (0.410–2.152)		
LDH level U/L	0.434	1.308 (0.615–3.099)		
Tumor size mm	0.108	0.526 (0.160–1.198)		
Ki-67 score	0.353	0.647(0.250–1.623)		
SUVbw * g/mL	0.122	1.844 (0.848–4.010)		
SUVlbm * g/mL	0.202	0.544 (0.213–1.388)		
SUVbsa * cm^2^/ml	0.178	1.770 (0.766–4.072)		
L-L SUV R *	0.253	1.602 (0.707–3.712)		
L-BP SUV R *	0.408	1.533 (0.557–4.232)		
gMTV * cm^3^	0.010	2.924 (1.288–6.635)	0.269	2.893 (0.157–4.234)
gTLG *	0.016	2.614 (1.194–5.726)	0.310	2.193 (0.057–4.444)
tMTV * cm^3^	0.001	4.383 (1.761–10.917)	0.023	1.228 (1.049–1.765)
TLG *	0.010	2.987 (1.291–6.913)	0.038	1.354 (1.069–1.989)
OS				
Gender	0.805	0.895 (0.373–2.150)		
Age years	0.202	0.544 (0.213–1.388)		
Tumor Stage	0.654	0.821 (0.347–1.943)		
B symptoms	0.127	2.193 (0.798–6.092)		
IPI score	0.319	0.629 (0.253–1.565)		
LDH level U/L	0.623	0.801 (0.331–1.940)		
Tumor size mm	0.108	0.526 (0.160–1.198)		
Ki-67 score	0.688	0.819 (0.313–2.166)		
SUVbw * g/mL	0.054	2.344 (0.983–5.588)		
SUVlbm * g/mL	0.343	0.603 (0.211–1.717)		
SUVbsa * cm^2^/mL	0.101	2.188 (0.857–5.585)		
L-L SUV R *	0.371	1.518 (0.607–3.796)		
L-BP SUV R *	0.355	1.723 (0.547–5.427)		
gMTV cm^3^	0.040	2.573 (1.043–6.344)	0.362	1.001 (0.998–1.004)
gTLG	0.022	2.768 (1.158–6.647)	0.375	0.597 (0.191–1.864)
tMTV * cm^3^	0.025	2.444 (1.943–4.338)	0.038	3.333 (1.201–8.858)
TLG *	0.022	2.896 (1.159–7.233)	0.026	3.161 (1.233–8.103)

HR: hazard ratio; CI: confidence interval; PFS: progression free survival; OS: overall survival; N°: number. * These features were dichotomized using thresholds values derived by ROC analysis, as showed in Table 1.

## Data Availability

All the data are available from the corresponding author upon reasonable request.

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
