# Peer review of "Prognostic Impact of Pretreatment 2-[18F]-FDG PET/CT Parameters in Primary Gastric DLBCL"

_medicina, 2021, doi:10.3390/medicina57050498_

Round 1
Reviewer 1 Report
None
Author Response
Dear Reviewer,
thank you very much for your positive evaluation.
Reviewer 2 Report
The study investigated the potential role of pretreatment PET/CT parameters in predicting treatment response and prognosis in primary gastric DLBCL. The description of the materials and methods is too long.
The suggestions of the first reviewer were taken into account, but it could be better to add the files with all the statistics.
Paragraph: 2.2 2-[18F]-FDG-PET/CT imaging and interpretation. Please modify: the matric reconstruction was 256x256 and the field of view 60….into the matrix ….
Please modify: Written consent was obtained before studies. Pre-treatment 2-[18F]-FDG-PET/CT scans were done within 7 days before starting, while 2-[18F]-FDG-PET/CT at the end of treatment, was executed at least 21 days after the completion of chemotherapy.
Please add to table 1 the IU to tMTV and gMTV expressed in cc (cm3), as done in table 3; the other parameters (SUV) can be considered unitless.
Please modify: In our study, we studied the potential prognostic role of several semiquantitative metabolic features (SUV-related variables and MTV and TLG) into In our study, we studied the potential prognostic role of several metabolic features (SUV-related variables and MTV and TLG)
Please modify: These evidences confirmed previous papers that underlined the superiority of MTV and TLG in comparison with SUV value in prognostication field.
In the discussion/conclusions, it might be useful to point out that the TLG and tMTV cutoff values could have a prognostic meaning no matter what the stage of gastric DLBCL.
Author Response
REVIEWER 2
The study investigated the potential role of pretreatment PET/CT parameters in predicting treatment response and prognosis in primary gastric DLBCL. The description of the materials and methods is too long.
The suggestions of the first reviewer were taken into account, but it could be better to add the files with all the statistics.
Paragraph: 2.2 2-[18F]-FDG-PET/CT imaging and interpretation. Please modify: the matric reconstruction was 256x256 and the field of view 60….into the matrix ….
Please modify: Written consent was obtained before studies. Pre-treatment 2-[18F]-FDG-PET/CT scans were done within 7 days before starting, while 2-[18F]-FDG-PET/CT at the end of treatment, was executed at least 21 days after the completion of chemotherapy.
Please add to table 1 the IU to tMTV and gMTV expressed in cc (cm3), as done in table 3; the other parameters (SUV) can be considered unitless.
Please modify: In our study, we studied the potential prognostic role of several semiquantitative metabolic features (SUV-related variables and MTV and TLG) into In our study, we studied the potential prognostic role of several metabolic features (SUV-related variables and MTV and TLG)
Please modify: These evidences confirmed previous papers that underlined the superiority of MTV and TLG in comparison with SUV value in prognostication field.
Thank you very much for your comments and observations. I have modified the parts you suggested.
In the discussion/conclusions, it might be useful to point out that the TLG and tMTV cutoff values could have a prognostic meaning no matter what the stage of gastric DLBCL.
About this point, I understand your suggestion and I agree with you. Thus, I have added a short paragraph to underline the role of MTV and TLg in comparison with stage of disease. "It seems interesting to note that metabolic volumetric parameters (MTV and TLG) have a prognostic value independently from stage of disease. The thresholds proposed by our work were 119 cm3 for tMTV and 852 for TLG; however, it is premature to suggest these cut-off values as universal because are strictly dependent to patients features and centre characteristics".
Reviewer 3 Report
The authors report results from a retrospective study regarding metabolic and volume-based parameters derived from FDG-PET in patients with gastric DLBCL. Different parameters are tested in regards to association with clinical outcome. The authors conclude that MTV and TLG are significantly associated with response to therapy, PFS and OS, whereas other metabolic parameters and clinicopathological features are not predictors of outcome.
The study is contemporary and of general interest to the field, but there are some issues to address. First of all the entire manuscript would benefit from language revision. There are several small grammatical errors, but also some sentences that make very little sense in their present form.
Specific comments:
Methods:
- The study includes patients from 2009-2020. Did you really use the same PET/CT scanner, protocol, reconstruction algorithm etc. for all the patients? And what reconstruction method was applied ? (OSEM, TOF…)
- Table 1 displays results from the study and should therefore be referred in the results section instead
- Regarding the use of Cox regression- have you investigated whether the assumptions of linearity and proportional hazards are fulfilled? And how?
Results
- Throughout the manuscript the terms “average” and “mean” are both used. Please choose one term, where I believe “mean” is the most widely adopted.
- Table 2: Why is TLG as the only variable stated as “mean (range)”, while the other have “SD” as descriptive term?
- Table 2: It seems like several of the variables have a skewed distribution, but in that case mean and SD are not completely appropriate for descriptive statistics. For instance some of the SD’s are larger than the mean value, which makes no sense in reality (not possible to have a negative gMTV). Have you considered log-transforming the data? Or at least using the median and percentile values instead?
- Table 3: Why do the SUV-variables have units in this table, but not in Table 2? You should be more consistent. And just a format issue in row 2 of the table “Incomplete response n 16” should be “Incomplete response (n=16)”
Discussion:
- Page 10, line 263-264. This sentence makes very little sense. Do you mean to encourage sharing data?
- Page 10, line 267 – “methodological and technical doubts are present” – can you elaborate which issues you are referring to?
Author Response
REVIEWER 3
The authors report results from a retrospective study regarding metabolic and volume-based parameters derived from FDG-PET in patients with gastric DLBCL. Different parameters are tested in regards to association with clinical outcome. The authors conclude that MTV and TLG are significantly associated with response to therapy, PFS and OS, whereas other metabolic parameters and clinicopathological features are not predictors of outcome.
The study is contemporary and of general interest to the field, but there are some issues to address. First of all the entire manuscript would benefit from language revision. There are several small grammatical errors, but also some sentences that make very little sense in their present form.
Dear Reviewer 3,
thank you very much for your comments and observations. I've tried to modify the paper following them. Moreover, I have revised the paper with the help of an English native speaker.
Specific comments:
Methods:
- The study includes patients from 2009-2020. Did you really use the same PET/CT scanner, protocol, reconstruction algorithm etc. for all the patients? And what reconstruction method was applied ? (OSEM, TOF…)
Yes, all the patients were acquired on the same scanner (DST GE). in my center, we have two PET scanners but to make the population homogeneous we preferred to include only patients from one scanner. also the other "technical" features like reconstruction algorithms, PET protocol are the same.
- Table 1 displays results from the study and should therefore be referred in the results section instead
I accept your suggestion to move Table 1 in Results sections. Thus, this table becomes Table 3 and so on
- Regarding the use of Cox regression- have you investigated whether the assumptions of linearity and proportional hazards are fulfilled? And how?
Ok I understand your points. Yes, "The assumption of proportionality of risks was tested using Schoenfeld residuals and the impact of outliers in the model fitness was analysed using martingale and score residuals. The data did not violate the proportional hazard assumption and the linearity was met".
Results
- Throughout the manuscript the terms “average” and “mean” are both used. Please choose one term, where I believe “mean” is the most widely adopted.
Sorry for the mistake. I have corrected as you indicated.
- Table 2: Why is TLG as the only variable stated as “mean (range)”, while the other have “SD” as descriptive term?
Sorry was an error. My fault. I have corrected it.
- Table 2: It seems like several of the variables have a skewed distribution, but in that case mean and SD are not completely appropriate for descriptive statistics. For instance some of the SD’s are larger than the mean value, which makes no sense in reality (not possible to have a negative gMTV). Have you considered log-transforming the data? Or at least using the median and percentile values instead?
Dear reviewer, thank you very much for this observation. I have revised our data and founded some errors for MTV SD values. So, I have changed these data.
- Table 3: Why do the SUV-variables have units in this table, but not in Table 2? You should be more consistent. And just a format issue in row 2 of the table “Incomplete response n 16” should be “Incomplete response (n=16)”
Sorry for these mistakes. I have added the units also in Table 1 and Table 2 and also (n=16).
Discussion:
- Page 10, line 263-264. This sentence makes very little sense. Do you mean to encourage sharing data?
Sorry for he unclear message. I have changed this phrase.
- Page 10, line 267 – “methodological and technical doubts are present” – can you elaborate which issues you are referring to?
I understand your doubt, thus I have clarified this point.
Round 2
Reviewer 3 Report
Thank you for the revised version. I think the manuscript has improved significantly. I just have a few minor issues left.
Methods:
- p. 4 line 134-137: 1) Martingale residuals are used to test linearity and not impact of outliers as you state. 2) Schoenfeld residuals are applied for continous variables, but your model includes variables that you have dichotomized - gMTV, gTLG, tMTV and TLG. Based on the Kaplan Meier plots in Figure 3 I am concerned about the assumption of proportional hazards in the dichotomous data, which is why I find it important to check thoroughly that the assumptions are met. So how did you examine proportional hazards for these variables?
Results:
- Page 4, line 148-152; the word "average" still appears alongside "mean". It would be more consistant to choose one term.
Otherwise I have no further comments.
Author Response
Reviewer 3
Thank you for the revised version. I think the manuscript has improved significantly. I just have a few minor issues left.
Methods:
- 4 line 134-137: 1) Martingale residuals are used to test linearity and not impact of outliers as you state. 2) Schoenfeld residuals are applied for continous variables, but your model includes variables that you have dichotomized - gMTV, gTLG, tMTV and TLG. Based on the Kaplan Meier plots in Figure 3 I am concerned about the assumption of proportional hazards in the dichotomous data, which is why I find it important to check thoroughly that the assumptions are met. So how did you examine proportional hazards for these variables?
Sorry for this mistake, I have clarified this part as you suggested.
Results:
- Page 4, line 148-152; the word "average" still appears alongside "mean". It would be more consistant to choose one term.
Sorry for this mistake, I have changed average with mean.